# Association of Acculturation Status with Longitudinal Changes in Health-Related Quality of Life—Results from a Cohort Study of Adults with Turkish Origin in Germany

**DOI:** 10.3390/ijerph18062827

**Published:** 2021-03-10

**Authors:** Lilian Krist, Christina Dornquast, Thomas Reinhold, Heiko Becher, Karl-Heinz Jöckel, Börge Schmidt, Sara Schramm, Katja Icke, Ina Danquah, Stefan N. Willich, Thomas Keil, Tilman Brand

**Affiliations:** 1Institute for Social Medicine, Epidemiology and Health Economics, Charité-Universitätsmedizin Berlin, 10117 Berlin, Germany; christina.dornquast@dzne.de (C.D.); thomas.reinhold@charite.de (T.R.); katja.icke@charite.de (K.I.); ina.danquah@uni-heidelberg.de (I.D.); stefan.willich@charite.de (S.N.W.); thomas.keil@charite.de (T.K.); 2German Center for Neurodegenerative Diseases (DZNE), 17489 Greifswald, Germany; 3Institute of Medical Biometry and Epidemiology, University Medical Center Hamburg-Eppendorf, 20246 Hamburg, Germany; h.becher@uke.de; 4Institute for Medical Informatics, Biometry und Epidemiology, University Hospital Essen, University Duisburg-Essen, 45122 Essen, Germany; k-h.joeckel@uk-essen.de (K.-H.J.); boerge.schmidt@uk-essen.de (B.S.); sara.schramm@uk-essen.de (S.S.); 5Institute of Global Health (HIGH), Heidelberg University Hospital, 69120 Heidelberg, Germany; 6Institute for Clinical Epidemiology and Biometry, University of Würzburg, 97070 Würzburg, Germany; 7State Institute of Health, Bavarian Health and Food Safety Authority, 97688 Bad Kissingen, Germany; 8Department of Prevention and Evaluation, Leibniz Institute for Prevention Research and Epidemiology—BIPS, 28359 Bremen, Germany; brand@leibniz-bips.de

**Keywords:** health-related quality of life, HRQL, acculturation, Turkish, migrants

## Abstract

Health-related quality of life (HRQL) among migrant populations can be associated with acculturation (i.e., the process of adopting, acquiring and adjusting to a new cultural environment). Since there is a lack of longitudinal studies, we aimed to describe HRQL changes among adults of Turkish descent living in Berlin and Essen, Germany, and their association with acculturation. Participants of a population-based study were recruited in 2012–2013 and reinvited six years later to complete a questionnaire. Acculturation was assessed at baseline using the Frankfurt acculturation scale (integration, assimilation, separation and marginalization). HRQL was assessed at baseline (SF-8) and at follow-up (SF-12) resulting in a physical (PCS) and mental (MCS) sum score. Associations with acculturation and HRQL were analyzed with linear regression models using a time-by-acculturation status interaction term. In the study 330 persons were included (65% women, mean age ± standard deviation 43.3 ± 11.8 years). Over the 6 years, MCS decreased, while PCS remained stable. While cross-sectional analyses showed associations of acculturation status with both MCS and PCS, temporal changes including the time interaction term did not reveal associations of baseline acculturation status with HRQL. When investigating HRQL in acculturation, more longitudinal studies are needed to take changes in both HRQL and acculturation status into account.

## 1. Introduction

Migrants are more often in poorer health than the population in the host country [1,2,3,4,5,6,7,8]. On the one hand this is due to migration-related factors such as the country of origin, reason for immigration, traumatic experiences or genetic dispositions to certain diseases [1,2,9,10]. On the other hand, low socioeconomic status, poorer education, cultural differences, language barriers and low health literacy can cause a poor health status by reduced access to health information and health services or a lower use of health screenings [1,11,12,13]. 

Many studies reported also lower subjective health-related quality of life (HRQL) among migrants than among the native population [14,15,16,17,18]. Reasons in addition to those mentioned may include environmental and migration-related factors, experienced discrimination, socioeconomic hardship, occupational stress and poor working conditions, but also acculturative stress [15,19,20,21,22,23,24].

Acculturation is an anthropological term that was first introduced in the late 19th century in the context of colonization, then, in the 1930s, defined for the use in studies including cultural terms such as attitudes, beliefs or values in the concept of acculturation [25,26]. In the 1960s, Gordon reconceptualized the term and described acculturation as a linear continuum ranging from not acculturated to acculturated [27]. Another concept was later developed by Berry, who described acculturation as a bidimensional construct with two coexisting components (adaption of the host culture and maintenance of the culture of origin) [28]. Berry’s model differentiates four acculturation strategies: (1) marginalization (low affiliation with both cultures); (2) separation (high origin-culture affiliation, low new-culture affiliation); (3) assimilation (high new-culture affiliation, low origin-culture affiliation) and (4) integration (high affiliation with both cultures) [28,29].

In the last years, a variety of studies have investigated acculturation as a predictor for physical or mental health in first and second generation migrants indicating that separation and marginalization rather predict poorer health outcomes than integration or assimilation [7,25,30,31]. A meta-analysis concluded that the most favorable acculturation strategy was integration, where migrants adapt to the host country but maintain their home country culture at the same time [21,32]. However, most studies in this field are cross-sectional and thus cannot provide information on the direction of the effect. It is as well possible that the respective acculturation strategy is rather the result of the experiences made in the host country, poor health status or low HRQL, than predicting these factors. Yoon et al. proposed that the effect of acculturation was mediated by other factors such as social connectedness and social status [22]. Additionally, the cultural distance between host country and country of origin may play a role resulting in greater difficulties of acculturation in non-Western or non-European migrants compared to Western migrants [33]. As acculturation is a complex concept, studies that examined HRQL in the context of acculturation show heterogeneous results. A cross-sectional study conducted in Greece found no association between acculturation and HRQL, but orientation to the heritage culture was negatively associated with psychological wellbeing [34]. Urzua et al. found that integration and separation were associated with more favorable quality of life in different domains in a migrant sample in Chile [35]. A cross-sectional study from Singapore showed a correlation between higher acculturation levels and higher HRQL [30]. Only two longitudinal studies investigated HRQL and acculturation. A study from the Netherlands showed that certain dimensions of acculturation were associated with higher HRQL among migrants living in the Netherlands [36]. A German study did not focus on acculturation, but showed a more pronounced decline of the mental component of the HRQL among 1st generation immigrants than among the host population, while the physical component was only associated with age [37]. Although a considerable number of studies on acculturation and HRQL among migrants have been carried out in recent years, there is still a lack of longitudinal data on this topic. In the 1960s and 1970s Germany recruited so called “guest workers” from predominantly Southern Europe and the Mediterranean region. Since then, persons with Turkish background have been the largest migrant group (currently 2.82 million) in Germany [38]. 

The aim of our study was therefore to investigate whether the acculturation status among persons of Turkish descent living in Germany has long-term effects on their health-related quality of life.

## 2. Materials and Methods

### 2.1. Study Sample and Design

For the present cohort study, 1236 adults of Turkish descent were recruited in two large German cities, Essen and Berlin. A detailed description of the baseline recruitment has been provided by Reiss et al. [39]. Briefly, the baseline assessment was conducted between 2012 and 2013 during the pretest phase of the German National Cohort Study (NAKO) with the aim to evaluate different recruitment strategies (register-based versus network approach) among persons with a Turkish background. For the first recruitment method, random samples from residents’ registration offices were drawn and an onomastic procedure was used to identify eligible persons. For the network approach, representatives of the Turkish community were contacted to spread information of the study and to support the recruitment. All recruited participants were invited to the study center where they completed a questionnaire and underwent some medical examinations (measurement of body height and weight, blood pressure and blood sample). In 2018–2019, all participants who had agreed to be recontacted were invited to the follow-up. Participants received a self-report questionnaire via mail asking for the health status, health behavior, HRQL and others. Baseline and follow-up recruitment were conducted using bilingual written invitations, telephone contacts, and home visits performed by bilingual study staff. A description of the follow-up recruitment and retention methods was published by Krist et al. [40]. The study was approved by the ethical review committee of the Charité—Universitätsmedizin Berlin (EA1/206/17), Germany, and registered at the German Clinical Trials Register under the registration number DRKS00013545. Written informed consent was obtained from all participants.

### 2.2. Measures

#### 2.2.1. Health-Related Quality of Life

For the assessment of HRQL, the Short Form Health Surveys 8 (SF-8) [41] and 12 (SF-12) [42,43] were used at baseline and at follow-up, respectively. Both instruments have been used in numerous countries and validated in different populations [44,45,46,47]. The instruments are short versions of the SF-36 yielding an eight-scale profile of different domains of physical and mental health (physical functioning, role participation with physical health problems (role–physical), bodily pain, general health, vitality, social functioning, role participation with emotional health problems (role–emotional) and mental health). For example, one question regarding mental health is “During the past 4 weeks, how much did personal or emotional problems keep you from doing your usual work, school or other daily activities?”. According to the SF-12 and SF-8 manual, single items are aggregated into a physical component summary score (PCS) and a mental component summary score (MCS). Summary scores were then transformed into T-scores with a mean of 50 and a standard deviation of 10 using US general population norms obtained from the QualityMetric 2009 Norming Study [43]. Internal consistency (Cronbach’s alpha) in our sample was α = 0.90 for the SF-8 and 0.89 for the SF-12. The correlation between SF-8 at baseline and SF-12 at follow-up was r = 0.47 (*p* > 0.001) for PCS and r = 0.45 (*p* > 0.001) for MCS.

#### 2.2.2. Acculturation

Acculturation status was assessed at baseline using the Frankfurt acculturation scale (FRACC) developed by Bongard et al. [48,49]. The scale consists of two subscales (subjectively assessed orientation towards the culture of origin (CO) and towards the host culture (HC)). Each scale includes ten items rated on a seven point Likert-like scale (0 = absolutely not, 6 = absolutely) assessing mechanisms of social integration: cultural identification, cultural practices and interethnic social networks. On each scale, the possible range was 0–60 points, with higher values indicating a stronger orientation to CO and HC, respectively. Cronbach’s alpha was acceptable, with α = 0.83 for CO and α = 0.78 for HC. Participants were categorized as having higher or lower orientation towards CO and HC, respectively, using the median of the subscales as cut-off. By combining the two subscales, four acculturation groups were created: integration (CO+, HC+), assimilation (CO−, HC+), separation (CO+, HC−) and marginalization (CO−, HC−) following the concept of Berry [28].

#### 2.2.3. Sociodemographic Covariates

We included sex, age, educational level (assessed at baseline) and net household income (assessed at follow-up) as sociodemographic variables (all variables were assessed via questionnaire). Age was categorized into five groups: 20–29 years, 30–39, 40–49, 50–59 and 60–69 years. Educational level was assessed as years of education, school type and country. We harmonized these data taking the Turkish schooling reform in 1997 into account [50] and then categorized them into <10 years, 10–12 years and >12 years of attained formal education in Turkey and/or Germany. Monthly net household income was categorized into <1000 Euro, 1000 to <2500 Euro and ≥2500 Euro. As additional migration-related variable, we included country of birth.

### 2.3. Statistical Analyses

The characteristics of the study population were analyzed descriptively by using frequencies (n) and means (±standard deviations). The association of HRQL with acculturation status was assessed in two steps. First, we analyzed the association of acculturation status (assessed at baseline) with each HRQL measurement using the cross-sectional data at baseline and at follow-up in ordinary linear regression models, respectively. In the second step, we used hierarchical linear regression models to assess change over time in HRQL. The advantage of this method is that all available data is included in the analysis while other methods such as analysis of variance with repeated measure include only subjects with complete datasets [51]. In the hierarchical linear model, baseline and follow-up observations were clustered in individuals. The relationship of acculturation status and change in HRQL was assessed by including a time (survey wave)-by-acculturation status interaction term. Since interaction terms are often difficult to interpret, we visualized the change over time in the acculturation groups by calculating predictive margins. Age, sex, education and income were included as covariates in all analyses. Assimilation was the reference category for acculturation status because it constituted the largest group.

Research on baseline data of this study indicated sex differences in HRQL (Brand et al. 2017). Therefore, we additionally applied sex-stratified models. Furthermore, since a large proportion of the participants had missing values on the FRACC scale (27%), we imputed the missing values in this variable using multiple imputations (MI, 5 imputations). MI uses various estimates to account for the uncertainty in the estimation of missing values. Compared to single imputations, MI yields wider standard errors and confidence intervals, which are supposed to be closer to the “true values” [52]. To ensure the consistency of the findings, we ran both a complete case analysis and an analysis with MI. All analyses were conducted using Stata 15 (StataCorp College Station, TX, USA).

## 3. Results

### 3.1. Characteristics of the Study Sample

Out of 1236 baseline participants, 1193 agreed to be recontacted. Of those, 330 completed the follow-up questionnaire (249/557 in Berlin and 81/636 in Essen). In Berlin, 248 persons refused actively or passively, while for 60 persons no valid address could be retrieved. In Essen, 544 persons refused actively or passively or could never be contacted, six persons died since the baseline observation.

Finally, 330 persons of Turkish origin were included in the analysis, but only 291 provided complete information for the HRQL scale at baseline and 314 at follow up (278 with complete information on both occasions, missing values were not imputed). The average age ± standard deviation was 43.3 ± 11.8 years at baseline. More women (65%) than men participated in the study and women were slightly overrepresented in the younger age groups (Table 1). About 21% of the study participants were born in Germany. Participants with their own migration experience lived for an average of 29.4 ± 10.6 years in Germany. A substantial share of the sample had a low level of education (37.4%) and a net household income of less than 1000 Euros (20.4%). While assimilation formed the largest category of the acculturation status in the total sample, this was only the case for men but not for women. There was a sex difference in distribution of the acculturation status with more women belonging to the separated group and more men being in the integrated group. Participants’ characteristics stratified for the acculturation status are presented in Appendix A.

### 3.2. Changes in HRQL

HRQL declined from baseline to follow-up in the total sample. In case of PCS, the decline was small (less than one scale point) and only observed among women. PCS sores slightly increased among men. The decline in MCS scores was larger and found for both women and men. Overall, there were large sex differences at baseline and follow-up on both HRQL scales with women consistently reporting lower scores (Table 1).

### 3.3. Association of Acculturation with HRQL

#### 3.3.1. Cross-Sectional Analysis

Analyzing the association between acculturation status and HRQL showed that while acculturation status was not related with PCS at baseline, PCS scores at follow-up were significantly lower in the integrated and the separated group when compared to the assimilated group. Conversely, MCS scores were significantly lower among the separated compared to the assimilated group at baseline, but no significant differences were observable at follow-up. Furthermore, the cross-sectional analysis confirmed the sex differences in both scales and showed a strong association between HRQL and level of income (Table 2). As the complete case and the MI analysis provided similar findings, we reported only the results from the MI analysis in Table 2.

#### 3.3.2. Longitudinal Analysis

As already indicated by the descriptive analysis, the hierarchical linear model showed a significant decline in MCS over time, whereas no substantial change occurred in the average PCS scores (Table 3). The “main effect” of the acculturation status in this model refers to the differences in PCS and MCS at baseline. It showed significantly lower MCS scores in the separated group compared to the assimilated group as already indicated by the cross-sectional analysis. The coefficients of the time-by-acculturation status interaction term indicate the difference in change over time in HRQL between the assimilated group and the other three acculturation groups. As shown in Table 3, no significant interaction effect occurred. This was also the case in the sex-stratified analysis (Appendix A).

Figure 1 and Figure 2 present a graphical depiction of the hierarchical linear model in Table 3. Figure 1 shows that PCS scores slightly increased among the assimilated while they decreased in the other groups. Thus, the difference in PCS scores increased, which corresponds to the negative coefficients in the interaction term (Table 3) and the significant differences in PCS at follow-up in the cross-sectional analysis (Table 2). However, the differences in MCS scores became smaller over time across the acculturation groups as can be seen in Figure 2. This corresponds with the positive coefficients in the interaction term for MCS in Table 3 and the lack of any significant difference in MCS at follow-up in the cross-sectional analysis (Table 2).

## 4. Discussion

### 4.1. Main Study Findings and Comparison with Other Studies

In the present cohort study among adults of Turkish descent living in Germany, HRQL was only partially associated with acculturation status. Separate cross-sectional analyses at baseline and follow-up revealed associations of acculturation with MCS only at baseline and with PCS only at follow-up. For both sum scores, the assimilated group had the best outcome, worst were separated and marginalized for PCS and MCS, respectively. When comparing our results to cross-sectional studies that used a similar definition of acculturation, it is noticeable that, in contrast, all of them found integration to be the most favorable acculturation strategy. However, only one focused on quality of life [35], while three investigated depression [7,8,32], and one anxiety [7]. Other studies focusing on quality of life but using other measures for acculturation showed that a higher acculturation score (measured with the “A Short Acculturation Scale for Filipino-Americans” (ASASFN)) [53], but also self-perceived integration [54], were associated with better quality of life. A reason for the better outcome in the assimilated group compared to the integrated group might be the higher percentage of second generation migrants among the assimilated. Although we controlled for age, education and income, we could not rule out that factors related to experiences and exposition before and shortly after migration (exposition to physically strenuous work and poor living condition) show a long-term effect here.

In our sample, MCS decreased from baseline to follow-up in all acculturation groups, while the differences between the groups remained relatively stable. Especially the marginalized and the separated had an MCS lower than the average indicating that those groups are particularly vulnerable. PCS decreased as well, except for the assimilated group that showed even an increase. The decline of MSC was, however, the steepest among the assimilated. At follow-up, both PCS and MCS were, however, still the highest in the assimilated group, followed by the marginalized, integrated and separated group. Considering the follow-up, the separated group had significantly lower MCS than the assimilated, while differences among the other groups were at a similar size, but were not statistically significant.

When taking the time as interaction term into account, our analyses revealed that no significant differences between the changes of HRQL in the acculturation groups occurred. The assimilated group with the highest baseline MCS score had even the most unfavorable trajectory compared to the other groups. A check for outliers did show a normal distribution, so there are rather other underlying reasons for that. One possible explanation could be that persons in that group were not successful with their acculturation strategy, e.g., they could have experienced discrimination, exclusion or did not have sufficiently social support, which can mitigate acculturative stress [15]. Reports of the Federal Antidiscrimination Agency confirm discrimination among persons with Turkish background [55,56]. One German study investigated time dependent changes in HRQL. Nesterko et al. showed a decline of MCS among migrants with Turkish background, and among second generation migrants compared to the German population and other migrant groups where no significant changes were observed [37]. This observation was confirmed by our study with the highest proportion of second generation migrants in the assimilated group (Appendix A). Similar to the results of Nesterko, the baseline MCS was better compared to the groups with a higher proportion of first generation migrants, but showed a stronger decrease over time. This might be explained with the difficult situation of this group torn between the culture of their parents (so that they are “stamped” as migrants by many Germans) and the German culture, their country of birth, which might create a distance to their Turkish community. This already challenging situation is often aggravated by discriminative experiences in the political, social or educational environment [57].

### 4.2. Strengths and Limitations

This study is to our knowledge the first longitudinal study examining trajectories of HRQL and their association with acculturation among a migrant sample in Germany. Second, the baseline recruitment was conducted very thoroughly and yielded a representative population-based sample of persons of Turkish descent covering different regions and a broad age range [39,58].

Some limitations have to be mentioned as well. Although great effort has been made to recruit participants for follow-up participation, the overall retention rate was rather low, which may have led to a selection bias. Second, our sample consists of first and second generation migrants, who were analyzed together, because the percentage of second generation migrants was too small (20.7%). Thus, the results may be valid rather for first than for second generation migrants. Lastly, acculturation status was assessed only at baseline so that changes over time could not be measured. However, first generation migrants of our sample lived on average for already almost 30 years in Germany. Therefore, the dynamic of acculturation might be negligible at this point, since significant changes have been reported only for the first years after migration [59,60].

### 4.3. Implications

The finding that acculturation patterns HRQL is also of importance for public health professionals, policy makers and researchers. Providing access to appropriate health care and preventative services on the one hand, but also inclusion in the host society as a whole is a task of the German state. Given the low HRQL among the separated and marginalized migrants, public health interventions should target these groups in order to improve their health. Peer to peer approaches such as the health initiative “With migrants for migrants (MiMi)” [61] are a promising way to improve reach and access to these groups. In addition, awareness by physicians and health professionals of the concept of acculturation should be enhanced in order to improve HRQL among migrants.

Although acculturation has been shown to be associated with HRQL in some way, more longitudinal studies should be conducted to assess trajectories of both acculturation and HRQL and their associations. Future research should also explore factors predicting different acculturation strategies in longitudinal studies in order to identify high risk profiles of migrants at an early stage. These risk profiles could then be used in the frame of screening programs or other health initiatives.

## 5. Conclusions

In a sample of adults of Turkish descent living in Germany, HRQL was partially associated with acculturation. While physical HRQL remained relatively stable, mental HRQL decreased significantly over six years in all four acculturation groups. Persons who adapted towards the host culture (assimilated) had the best HRQL at baseline and follow-up. There was no association of acculturation with HRQL changes over time. 

Further research should include longitudinal studies with larger study samples and assess acculturation status at each follow-up to take its dynamic nature into account. This may be especially useful among recently arrived migrants.

## Figures and Tables

**Figure 1 ijerph-18-02827-f001:**
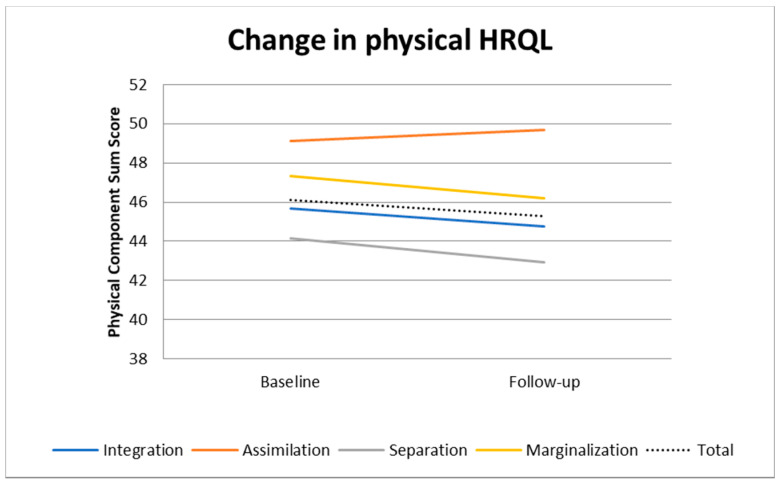
Change over time in physical component summary score (PCS) by acculturation status, adjusted for age, sex, education and income (predictive margins).

**Figure 2 ijerph-18-02827-f002:**
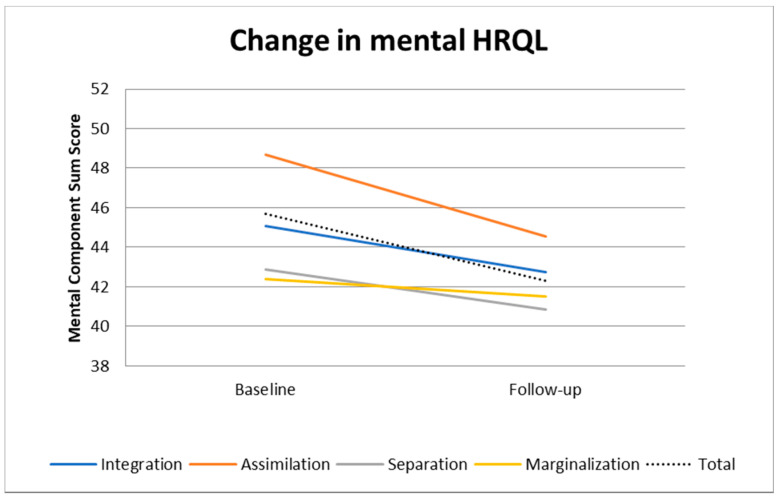
Change over time in mental component summary score (MCS) by acculturation status, adjusted for age, sex, education and income (predictive margins).

**Table 1 ijerph-18-02827-t001:** Characteristics of study participants.

N	Men (*n* = 118)	Women (*n* = 211)	Total (*n* = 330) ^§^
***Baseline variables***			
**Age groups** (%)			
20–29	9.3	19.0	15.5
30–39	20.3	26.1	24.0
40–49	39.0	28.0	31.9
50–59	19.5	14.7	16.4
60–69	11.9	12.3	12.2
**Country of birth** (%)			
Turkey	79.7	70.6	73.9
Germany	13.6	24.6	20.7
Missing	6.8	4.7	5.5
**Educational level** (%)			
Low	30.0	17.0	37.4
Medium	29.7	35.6	26.5
High	31.2	42.4	25.1
Missing	9.1	5.1	10.9
**Monthly net income** (%)			
<1000 Euro	16.1	22.8	20.4
1000–<2500 Euro	44.9	41.2	42.6
2500 Euro or more	29.7	20.9	24.0
Missing	9.3	15.2	13.1
**Acculturation status** (%)			
Integration	20.3	11.9	14.9
Assimilation	33.1	21.3	25.5
Separation	17.8	23.7	21.6
Marginalization	11.0	10.4	10.6
Missing	17.8	32.7	27.4
**PCS** (mean, SD)	47.6 (9.1)	45.2 (10.2)	46.1 (9.8)
**MCS** (mean, SD)	49.0 (10.4)	43.7 (11.0)	45.7 (11.0)
***Follow-up variables***			
**PCS (mean, SD)**	48.4 (9.3)	43.6 (10.5)	45.3 (10.3)
**MCS (mean, SD)**	45.8 (9.8)	40.5 (10.5)	42.3 (10.6)

Numbers are percentages or means (SD); SD: standard deviation; PCS: physical component summary score; MCS: mental component summary score; ^§^ missing information on sex in one case.

**Table 2 ijerph-18-02827-t002:** Cross-sectional analysis of the association between acculturation status and health-related quality of life at baseline and follow-up (linear regression).

	PCS Baseline	PCS Follow-Up	MCS Baseline	MCS Follow-Up
**N**	287	289	287	289
**Acculturation status** (Ref. Assimilation)				
Integration	−1.86	−3.95 *	−2.96	−1.87
	[−5.69, 1.98]	[−7.37, −0.54]	[−6.59, 0.66]	[−5.89, 2.16]
Separation	−3.44	−3.40 *	−4.45 *	−2.87
	[−7.50, 0.61]	[−6.85, −0.04]	[−8.57, −0.34]	[−638, 0.63]
Marginalization	−0.88	−1.92	−2.90	−1.81
	[−4.65, 2.87]	[−5.27, 1.43]	[−7.98, 2.17]	[−6.62, 2.98]
**Sex** (Ref. Men)				
Women	−1.60	−3.89 **	−4.08 **	−3.51 **
	[−4.10, 0.89]	[−6.20, −1.57]	[−6.87, −1.30]	[−6.05, −0.97]
**Monthly net income** (Ref. <1000€)				
1000–<2500€	2.01	2.52	3.84 *	2.29
	[−1.10, 5.12]	[−0.42, 5.47]	[0.33, 7.36]	[0.03, 6.25]
≥2500€	4.65 **	6.31 ***	7.16 ***	6.97 ***
	[1.19, 8.12]	[3.01, 9.61]	[3.21, 11.1]	[3.30, 10.64]
**Educational level** (Ref. Low)				
Medium	0.06	1.30	−0.56	1.17
	[−2.89, 3.00]	[−1.47, 4.07]	[−3.84, 2.72]	[−1.93, 4.27]
High	2.52	1.97	0.74	2.51
	[−0.48, 5.48]	[−0.85, 4.78]	[−2.62, 4.11]	[−0.62, 5.65]

Adjusted for age; 95% confidence intervals in brackets; * *p* < 0.05, ** *p* < 0.01, *** *p* < 0.001; PCS: physical component summary score; MCS: mental component summary score.

**Table 3 ijerph-18-02827-t003:** Change over time in health-related quality of life and acculturation status (hierarchical linear regression).

	PCS		MCS	
**N**	585		585	
	Coef.	[95% CI]	Coef.	[95% CI]
**Change over time**	0.24	[−1.94, 2.41]	−3.86 **	[−6.21, −1.51]
**Acculturation status** (Ref. Assimilation)				
Integration	−1.91	[−5.65, 1.82]	−3.13	[−6.63, 0.35]
Separation	−3.02	[−6.87, 0.83]	−4.04 *	[−8.00, −0.08]
Marginalization	−1.03	[−4.71, 2.63]	−2.37	[−7.27, 2.51]
**Time by acculturation** (Ref. Time#Assimilation)				
Time#Integration	−2.04	[−6.68, 2.61]	1.44	[−3.24, 6.13]
Time#Separation	−0.86	[−4.16, 2.45]	1.06	[−2.41, 4.53]
Time#Marginalization	−0.93	[−5.06, 3.19]	0.27	[−4.11, 4.65]

Adjusted for age, sex, education and income; 95% confidence intervals in brackets; * *p* < 0.05, ** *p* < 0.01; PCS: physical component summary score; MCS: mental component summary score.

## Data Availability

The data presented in this study are available on request from the corresponding author.

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
