# Peer review of "Association of Acculturation Status with Longitudinal Changes in Health-Related Quality of Life—Results from a Cohort Study of Adults with Turkish Origin in Germany"

_ijerph, 2021, doi:10.3390/ijerph18062827_

Round 1
Reviewer 1 Report
Thank you for the opportunity to review this article. The time involved in submitting your manuscript is greatly appreciated.
Despite this, the article presents a series of issues that must be noted and mended. The recommendations are presented separately by sections. Hopefully, they would be useful.
Title: the title does not adequately reflect the content of the paper. Please, try to change it to better inform the readers about the relationships between the variables that you test .
Abstract:
Less information appears in the abstract. Maybe expanded by adding the most relevant findings. Please, take into account that the abstract is the unique part of your paper that most of the readers could read. Hence, more information would be better.
Introduction:
Firstly, some of the references that you cite are too old, in example, Berry 1997. Even though the most relevant studies should be referenced, also the RECENT research must be included.
At the end of the literature review, the aims and the questions in the research should appear. Maybe to formulate the questions as a hypothesis would be an option to clear this aspect. Another commentary, it is the possibility of including this part at the final of the introduction part; even a separate section could be a good option, in order to clear the final of the introduction and to serve as a connection with the method.
Method:
Please, try to better describe the sociodemographic data of your participants. In the same sense, give the readers with detailed information about the procedure for recruiting participants and collecting data.
Related to the instruments, please better inform about their psychometric quality and give to the readers some examples of the items. If you can, please inform me about previous studies where the same instrument has been used and the reliability obtained in that research.
Data analyses
Please, explain to the readers which procedures of statistical analyses have been used and justify your decisions.
Results
The results should be presented following the same order as the introduction and hypotheses. Also, the same order must be used in the Tables. This simplifies the work for readers.
Finally, the repetition is constant all over the article. Please, try to change the words in order to do the reading more interesting and motivating
Discussion:
First of all, try to better adjust your conclusions to the findings. Or to say in other words, please try to justify more clearly the connection between your conclusions and your findings.
Finally, a section related to limitations, future lines of investigations and the principal contributions of the research could be interesting. Your paper has a lot of relevant implications for society and policymakers, but you need to elaborate more on this topic.
Conclusion:
They don’t appear new conclusions on this part. This part does not add any new to the rest of the paper. Please, try to condense your findings, or to highlight your main contribution to the field.
Reviewer 2 Report
See attached file.

Round 2
Reviewer 2 Report
Thanks for the edits. I find all my concerns addressed and the manuscript seems a lot improved and I recommend it for publication. Please go over the grammar. I still see some minor spelling mistakes.
Thanks and good luck!